# Improving routine childhood immunisation outcomes in low-income and middle-income countries: an evidence gap map

Mark Engelbert ![ORCID],[1] Monica Jain ![ORCID],[2] Avantika Bagai,[3] Shradha S Parsekar ![ORCID] [4]

[1]International Initiative for Impact Evaluation, London, UK
[2]International Initiative for Impact Evaluation, New Delhi, India
[3]Development Solutions, New Delhi, India
[4]Public Health Evidence South Asia, Prasanna School of Public Health, Manipal Academy of Higher Education, Manipal, Karnataka, India

**Correspondence to**
Mark Engelbert,
mengelbert@3ieimpact.org

## ABSTRACT

**Objective** To support evidence-informed decision-making, we created an evidence gap map to characterise the evidence base on the effectiveness of interventions in improving routine childhood immunisation outcomes in low-income and middle-income countries (LMICs).

**Methods** We developed an intervention–outcome matrix with 38 interventions and 43 outcomes. We searched academic databases and grey literature sources for relevant impact evaluations (IEs) and systematic reviews (SRs). Search results were screened on title/abstract. Those included on title/abstract were retrieved for full review. Studies meeting the eligibility criteria were included and data were extracted for each included study. All screening and data extraction was done by two independent reviewers. We analysed these data to identify trends in the geographic distribution of evidence, the concentration of evidence across intervention and outcome categories, and attention to vulnerable populations in the literature.

**Results** We identified 309 studies, comprising 226 completed IEs, 58 completed SRs, 24 ongoing IEs and 1 ongoing SR. Evidence from IEs is heavily concentrated in a handful of countries in sub-Saharan Africa and South Asia. Among interventions, the most frequently evaluated are those related to education and material incentives for caregivers or health workers. There are gaps in the study of non-material incentives and outreach to vulnerable populations. Among outcomes, those related to vaccine coverage and health are well covered. However, evidence on intermediate outcomes related to health system capacity or barriers faced by caregivers is much more limited.

**Conclusions** There is valuable evidence available to decision-makers for use in identifying and deploying effective strategies to increase routine immunisation in LMICs. However, additional research is needed to address gaps in the evidence base.

## STRENGTHS AND LIMITATIONS OF THIS STUDY

⇒ We adopt a rigorous and systematic approach to literature mapping.
⇒ We map a broad swath of the literature on routine childhood immunisation in low-income and middle-income countries.
⇒ Our analysis is complemented by an interactive online map.
⇒ Coverage of intermediate outcomes (such as those related to health system capacity) in this evidence gap map may be incomplete if studies on these outcomes do not target routine vaccination specifically.

## INTRODUCTION

Immunisation remains one of the most cost-effective interventions to prevent and control life-threatening infectious diseases. A 2016 study found that for every US dollar invested in vaccination in the world's 94 lowest-income countries, US$16 can be saved in healthcare costs, lost wages and lost productivity due to illness and death.[1] Despite these benefits, rates of routine immunisation of children remain below targets. In 2019, only 85% of children worldwide received the full course of the diphtheria, tetanus and pertussis (DTP) vaccine, leaving nearly 20 million children without full protection against these diseases.[2] Even with adequate vaccine availability, national immunisation systems in many low-income and middle-income countries (LMICs) are unable to achieve high immunisation coverage, which requires both reliable health service delivery and attenuation of behavioural, social and practical constraints faced by caregivers. Many LMICs struggle with last-mile connectivity, limited supply of health personnel, difficulty training health personnel with relatively low literacy/skill levels and lack of reliable monitoring and surveillance systems to provide reliable up-to-date data to improve targeting of immunisation programmes.[3] Behavioural, social and practical constraints faced by caregivers include caregivers' and communities' concerns about vaccine safety, lack of knowledge about the recommended schedule, fear of adverse events following immunisation,

poor quality and reliability of immunisation services and difficulty accessing health services.[4]

In the last few decades, interventions have emerged specifically aimed at strengthening national-level immunisation programmes by addressing delivery constraints. However, even with strengthened routine immunisation programmes, marginalised and vulnerable communities are at risk of being overlooked. Thus, a number of international and national policy frameworks are increasingly focusing on multipronged approaches that provide contextualised solutions to address constraints in delivery of immunisation services as well as behavioural, social and practical barriers faced by caregivers.

This study presents an evidence gap map (EGM) of impact evaluations (IEs) and systematic reviews (SRs) measuring the effects of interventions on outcomes related to routine child immunisation. An EGM aims to establish what is known and unknown about an evidence base in a thematic area.[5] A map is populated by employing a systematic search and screening process to identify all relevant IEs and SRs, completed or ongoing, that meet a set of prespecified inclusion criteria. All studies that met these criteria are mapped onto a matrix of interventions and outcomes and displayed in a grid-like framework on an interactive platform.

EGMs highlight both absolute gaps (an empty cell in the framework), which could be filled with new IEs, and synthesis gaps (multiple IEs but no SRs), which could be filled with evidence synthesis. Furthermore, they can be used to highlight potentially over-researched areas, where the effects of a particular intervention have been evaluated against a range of outcomes (or vice versa) and where additional studies may not be necessary.

The objectives of this EGM are to:
1. Present a framework of types of interventions and outcomes related to routine child immunisation in LMICs.
2. Using this framework, map available SRs and IEs by their interventions and outcomes, with a summary of findings provided in this paper.
3. Assess the quality of the SRs falling under the framework.

## SCOPE
### Conceptual framework
Drawing on the literature on routine child immunisation in LMICs, we developed a theory of change in which we identified key factors—from the perspectives of caregivers and communities, vaccinators, health systems and beyond—that influence whether children receive the full course of routine vaccinations.[6–8] These factors form the basis for our intervention–outcome matrix.

The theory of change suggests how interventions may be oriented to address afore-mentioned key factors and lead to greater immunisation coverage and better health outcomes (figure 1). It suggests that interventions directed at caregivers and communities create greater desire, motivation, awareness, opportunity and decision-making power to pursue vaccination services (behavioural, social and practical factors). Interventions aimed at health providers and systems increase their capacity and accountability to deliver quality and timely

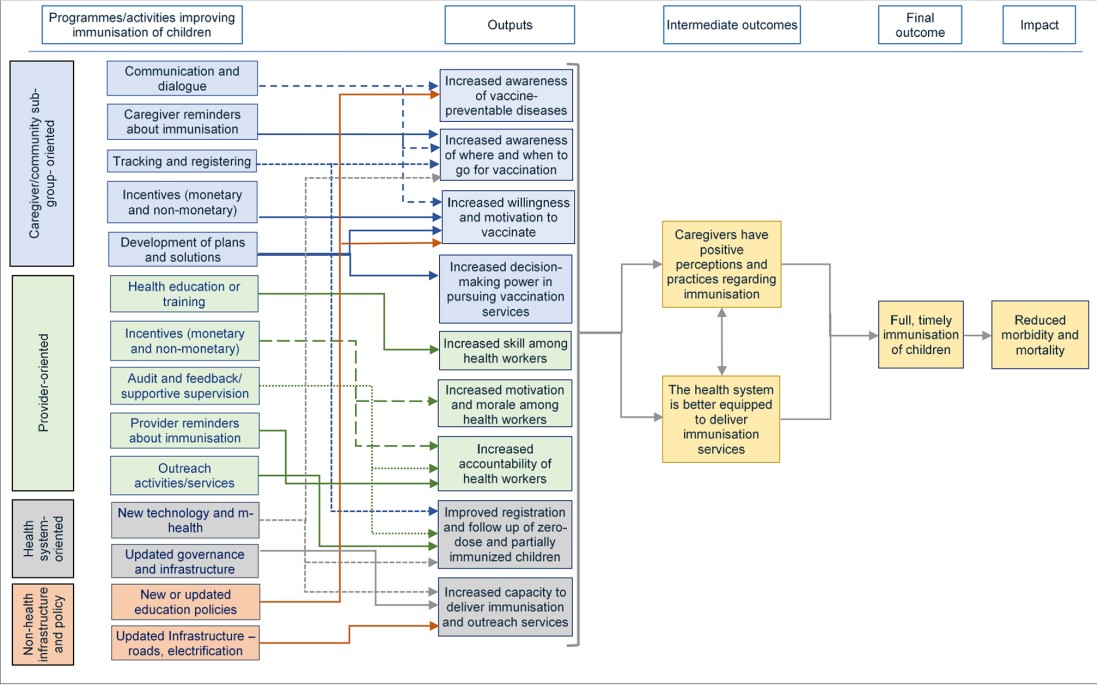

Note: certain lines are dashed to increase the readability of the diagram; the different line styles do not indicate different types of relationships

**Figure 1** Theory of change. Note: certain lines are dashed to increase the readability of the diagram; the different line styles do not indicate different types of relationships.

vaccination services. Interventions addressing non-health barriers such as poor road infrastructure facilitate better delivery of vaccination services. Addressing behavioural, social and practical barriers faced by caregivers and communities, along with improved service delivery, increase the number of children who receive all vaccine doses on time. Full and timely immunisation coverage, in turn, reduces morbidity and mortality in the population.

## METHODS

### Inclusion and exclusion criteria

We included IEs and SRs evaluating the effectiveness of interventions to increase immunisation of children under five in LMICs. Interventions did not have to be expressly designed for the purpose of increasing immunisation rates; rather, we searched for all IEs and SRs that provided evidence on the impact of any intervention on routine immunisation coverage, morbidity or mortality from vaccine-preventable diseases, or any intermediate outcome that precedes them in the theoretical causal chain. There were no restrictions by publication date, publication type/status or language.

The EGM includes experimental and quasiexperimental IEs that estimate the causal impact of an intervention, as compared with usual practice or an alternative treatment, by establishing a counterfactual.[9]

### Intervention–outcome framework development

To develop the intervention–outcome framework for this EGM, we developed an initial matrix by drawing on the research literature on determinants and constraints related to vaccination in LMICs,[6 10–13] as well as UN agency reports.[14 15] We requested feedback on the draft framework from an advisory group of research and policy experts.

This resulted in a framework with 38 unique interventions and 43 unique outcomes. These are each grouped together in a three-level hierarchy of categories. The interventions are grouped broadly by who or what is targeted: caregivers, the health system, non-caregiver community members, the community as a whole or policies and institutions beyond the health system. Our outcome taxonomy has three levels. At the highest level, we divide outcomes into three broad types: (a) those related to behavioural, social and practical barriers faced by caregivers and communities, (b) those related to delivery of vaccination services by formal and informal components of the health system and (c) those related to vaccination coverage/timeliness and health outcomes (ie, morbidity and mortality from vaccine-preventable diseases). Within each of these types are a series of mid-level (broad) categories, and within each of these are specific outcome categories. The inclusion of intermediate outcomes in (a) and (b) allows us to assess the extent to which the evidence base contains insights not only on the effectiveness of interventions in improving vaccination coverage/timeliness and health outcomes but also the effectiveness

**Figure 2** Broad categories in the intervention–outcome matrix.

of interventions in improving their important causal antecedents. The outcomes in category (c) are further divided into more specific categories corresponding to various prerequisites for universal vaccination coverage. The framework also includes seven 'cross-cutting themes' that capture possible subgroup analyses (eg, by sex of the child or maternal education, socioeconomic characteristics or hard-to-reach populations) as well as other key features of interest (eg, whether the study presents any form of cost analysis). A sketch of the matrix, showing the first-level and second-level intervention and outcome categories, appears in figure 2 below, while the full framework, along with definitions for each category, can be found online (online supplemental appendix A).

Many studies evaluated interventions that included multiple components (eg, training for health workers (HWs) and sensitisation meetings with women). We determined that the best approach was to capture all major intervention components for each study. That is, some studies have multiple intervention codes applied to them. In this way, the EGM can be considered a map of 'instances of evidence' rather than a map of studies. A study provides an 'instance of evidence' when it provides evidence about the impact of a particular type of activity on a particular outcome. Because many studies assess multiple interventions and/or multiple outcomes, many studies provide multiple instances of evidence. While mapping intervention components separately has limitations (see Limitations section below), it is well suited to a literature like that covered in this EGM, where multicomponent interventions are common, but there are few 'standard packages' of intervention components that are frequently implemented and evaluated together. This approach also has the advantage of allowing users of the map to easily identify studies that evaluate particular intervention components of interest.

### Search and screening

This EGM employed systematic methods for search, screening and data extraction, following International Initiative for Impact Evaluation recommended guidelines for EGMs.[5 16] In consultation with an information specialist, we designed a systematic search strategy

covering academic databases (online supplemental appendix B.1) and sources of the 'grey literature', that is, institutional websites (online supplemental appendix B.2). Database searches were conducted on 17 May 2019, and an updated search was completed on 5 May 2020 to identify records that had been published while the EGM was ongoing. Search results were imported into EPPI-Reviewer V.4 and deduplicated. We also performed backward and forward citation tracking on all included studies (using Google Scholar for the latter). Finally, we contacted experts in the field to identify additional studies.

All search results were screened on title/abstract by a team of trained reviewers. Two reviewers independently screened each abstract. Studies included at the title/abstract level were retrieved for full review. Full texts were also independently screened by two reviewers. All screening disagreements were resolved through discussion.

### Data extraction and analysis

Data extraction was conducted independently by two trained coders using a standard template (online supplemental appendix B.3). For each study, we extracted data about the location of the study, the analysis methods used, interventions and outcomes and other characteristics of the population studied. All data were extracted and analysed using Microsoft Excel.

When our search identified multiple publications resulting from the same study (eg, a working paper and a journal article), we counted these as a single study and data extraction for each study drew on all the publications identified for that study.

When studies evaluated interventions with multiple components, we coded each intervention separately. While this approach captures important detail about the intervention activities that have been included in IEs, it can also overstate the apparent amount of evidence by 'double-counting' studies with multicomponent interventions. Hence, we also conducted an analysis with all multicomponent studies grouped into a single 'multicomponent' category, rather than having each component coded individually.

For SRs, we conducted critical appraisals using a standardised tool for assessing the quality and systematicity of each review's search, screening, data extraction and analysis methods (online supplemental appendix B.4). Based on these appraisals, we assigned each review a confidence rating of high, medium or low.

### Patient and public involvement

As the unit of analysis for this EGM is research studies rather than human subjects, there was not a need to involve individual patients. However, to ensure the relevance of our research to a broad array of stakeholders, we engaged an advisory group comprising immunisation experts from the research, practitioner and policy-maker domains. This group advised on the development of our

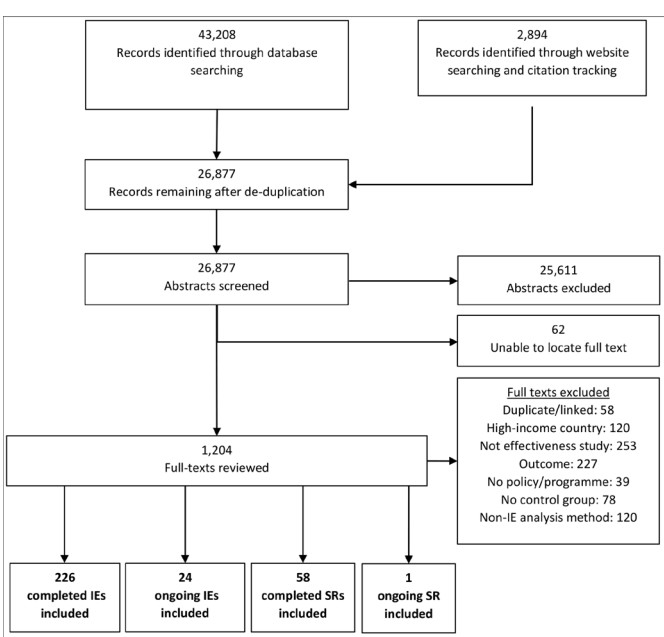

**Figure 3** Preferred Reporting Items for Systematic Reviews and Meta-Analyses diagram of search and screening process. IE, impact evaluation; SR, systematic review.

framework, search strategy, data analysis and dissemination plans.

## FINDINGS

Figure 3 presents the results of the search and screening process. Our search retrieved 46 102 records. After deduplication and screening, we identified 323 studies that assess interventions to increase routine immunisation rates in LMICs. These include: 236 completed IEs, 24 ongoing IEs, 58 completed SRs and 1 ongoing SR (online supplemental appendix C).

### Geographic distribution of the IE evidence base

A total of 85 countries are represented among the IEs included in the EGM. The regions most commonly represented are sub-Saharan Africa (95 IEs) and South Asia (75 IEs). There is comparatively little evidence from Europe and Central Asia (five IEs) or the Middle East and North Africa (three IEs), with somewhat greater numbers in the East Asia/Pacific and Latin America/Caribbean regions. India is by far the most represented country with 45 IEs. Other South Asian countries are also well represented, including Pakistan (12 IEs) and Bangladesh (9 IEs). Evidence is fairly well-distributed across sub-Saharan Africa, though with some notable gaps. Nigeria is the most represented country with 14 IEs, followed by Kenya (12 IEs), Rwanda (6 IEs) and Ethiopia (7 IEs).

Figure 4 compares the amount of IE evidence with each country's DTP3 vaccine coverage rate in 2019, according to WHO data.[17] There are several countries that have both very low coverage rates and limited evidence available. Countries with very low coverage rates, but that have not been the site of any IEs, include Papua New Guinea (35%

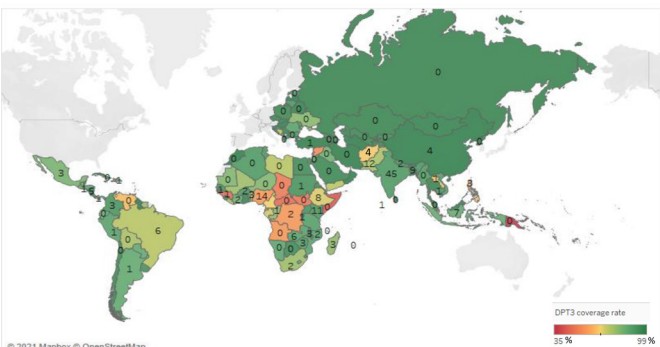

**Figure 4** Impact evaluations by country diphtheria, tetanus and pertussis (DPT)3 coverage rate. Map based on longitude (generated) and latitude (generated). Colour shows sum of DPT3 coverage rate. The marks are labelled by sum of impact evaluations. Details are shown for country.

coverage rate), Somalia (42%), Central African Republic (47%), Guinea (47%) and South Sudan (49%). Nigeria presents a rare case of a country with very low coverage (57%) that has built a considerable evidence base with 14 IEs.

### Interventions and outcomes covered in the evidence base
#### Intervention coverage

The intervention type most commonly evaluated is caregiver incentives and motivation, with 79 IEs and 35 SRs (figure 5) (Because some studies evaluate multiple interventions in this category, summing the studies in each intervention (shown in figure 5) yields numbers greater than 79 IEs and 35 SRs). Interventions involving changes to health system governance, policies and financing are also frequently evaluated (54 IEs and 20 SRs). In general, interventions providing education, training or incentives to either caregivers or HWs are among the most commonly evaluated. Although interventions providing knowledge and education to caregivers are frequently evaluated, only four IEs have explicit components aimed to dispel misinformation or misconception regarding vaccination, and only one SR touches on it. Among interventions leveraging community engagement, there is a concentration of evidence on the use of community dialogue for increasing immunisation coverage (26 IEs and 15 SRs).

The most commonly examined specific interventions are material/monetary incentives for caregivers (43 IEs, 19 SRs), health system strategic planning (33 IEs, 14 SRs) and written or pictorial messages for caregivers (29 IEs, 25 SRs), which includes SMS reminders or motivational messages. There is comparatively little work on non-material incentives for caregivers (3 IEs, 1 SR), which includes, for example, community recognition as an incentive to vaccinate. Among interventions to motivate HWs, pay-for-performance schemes are by far the most commonly evaluated (27 IEs, 14 SRs).

Several evidence gaps are evident. Apart from monetary incentives and pay-for-performance schemes, there is very little evidence on incentives, motivation and informational messages for HWs. There are also significant gaps in outreach to migrant populations (one IE, no SRs) and campaigns to vaccinate refugee populations (no IEs or SRs). We did not identify any IEs or SRs with interventions framed specifically as outreach to vaccine-hesitant groups, although there is some evidence on outreach to vulnerable populations (10 IEs, 8 SRs), which may overlap with vaccine-hesitant groups to some extent. Cold chain infrastructure improvements present another notable gap, with only two IEs and one low-confidence SR evaluating such interventions. There is a potential synthesis gap with respect to community HW training and education, for which there are 22 IEs but no medium-confidence or high-confidence SRs.

Of the 226 IEs in our EGM, 87 evaluated multicomponent interventions (online supplemental appendix D, figure D.1). We find that certain interventions are very frequently evaluated as parts of multicomponent interventions. These include written or pictorial materials for caregivers, which appears in 29 studies when intervention components are counted separately, but has been evaluated 15 times as a standalone intervention. Similarly, health system strategic planning appears in 33 studies but has been evaluated as a standalone intervention 9 times. Furthermore, interventions that train formal or informal HWs are seldom evaluated independently; rather, most of the time they appear in bundles of interventions that are evaluated together.

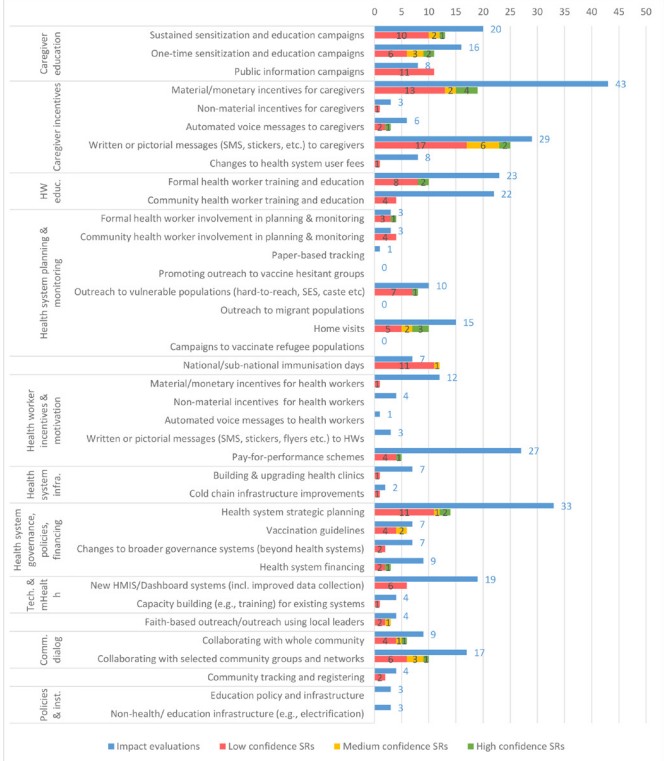

**Figure 5** Impact evaluations and systematic reviews (SRs) by specific intervention category. HW, health worker; SES, socioeconomic status; HMIS, health management information system.

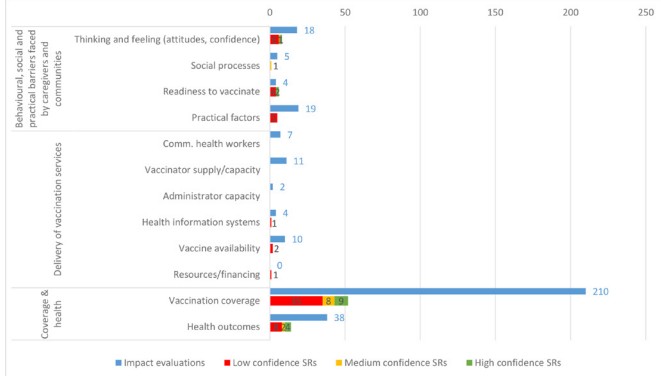

**Figure 6** Impact evaluations and systematic reviews (SRs) by broad outcome category.

## Outcome coverage

By far, the most commonly measured outcome categories fall under vaccination coverage, with 210 IEs and 52 SRs examining 1 of these outcomes (figure 6). Health outcomes are also frequently measured, included in 38 IEs and 14 SRs. In contrast, there is comparatively little evidence on health systems' capacity to deliver vaccination services. There is a particular gap regarding the effects of interventions on the health system's resources and financing for vaccination services—we found no IEs and one low-confidence SR that measured this outcome (We reviewed this SR 18to determine whether the studies it reviewed would qualify as IEs that could be included in our map. However, none of the studies on *Resources and financing* in this review met our methodological inclusion criteria). A potential synthesis gap exists on community HW supply and capacity, for which we identified seven IEs but no SRs.

Regarding outcomes related to behavioural, social and practical barriers faced by caregivers and communities, some outcome categories are relatively well studied, while others have very little evidence. We identified 18 IEs and 8 SRs that considered caregivers' attitudes about immunisation. The practical factors that influence caregivers' opportunities to vaccinate (eg, their knowledge of how to access services, the costs associated with vaccinating, or their perception of side effects) are also comparatively well studied, with 19 IEs and 5 SRs (although these SRs are all low confidence). In contrast, few studies examine the social processes affecting caregivers' vaccination decisions (five IEs, one medium-confidence SR) or their readiness to vaccinate (four IEs, six SRs).

Only two specific outcome categories on the caregiver side of our framework were examined in more than 10 studies: caregivers' knowledge about immunisation (16 IEs, 5 SRs) and retention of vaccination cards (12 IEs, 2 SRs) (figure 7). Evidence is much less consistently available in the practical factors category, where we identified one study each assessing impact on the actual cost and the perceived convenience of vaccinating children.

Relatively little evidence examines specific outcomes related to delivery of immunisation services (figure 8).

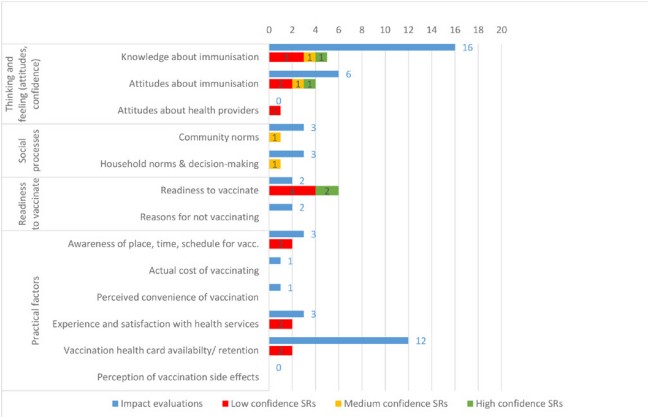

**Figure 7** Impact evaluations and systematic reviews (SRs) by specific outcomes related to behavioural, social and practical barriers faced by caregivers and communities.

The two most commonly examined outcomes are formal HW motivation, capacity and performance (10 IEs) and vaccine stockouts (8 IEs). These also present potential synthesis gaps, as we found no SRs covering HW motivation, and only two low-confidence SRs covering vaccine stockouts.

Vaccination coverage and child health outcomes are fairly well studied and there are limited evidence gaps or synthesis gaps (figure 9). One gap, however, concerns coverage of inactivated polio vaccine (IPV), which was assessed in one IE and one SR. As countries begin to incorporate IPV into their vaccination schedules in line with WHO recommendations,[18] it will be important to close this evidence gap. Another gap is related to coverage of zero dose or unvaccinated children, which is an important target group for WHO's Immunization Agenda 2030[19] but has been assessed in only 10 IEs and 1 SR.

Our map identified much more evidence on vaccination coverage than on downstream health outcomes. This pattern likely appears in our map because we only included studies measuring morbidity or mortality if they also measured vaccination coverage—that is, if they provided data that could be used to assess whether the intervention affected morbidity and mortality via

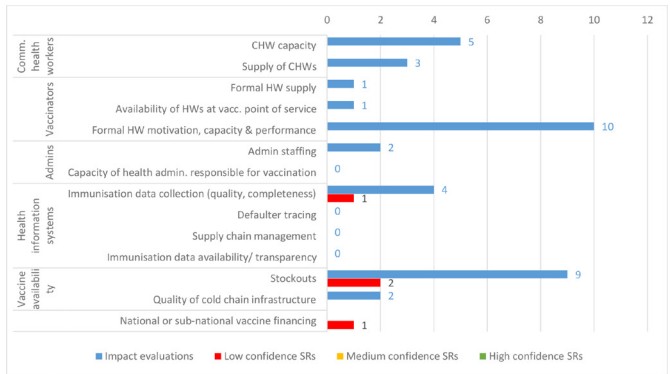

**Figure 8** Impact evaluations and systematic reviews (SRs) by specific outcomes related to delivery of vaccination services. HW, health worker; CHW, community health worker.

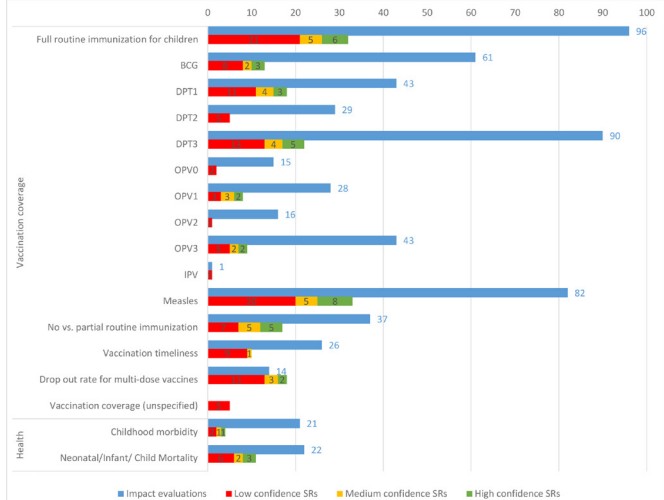

**Figure 9** Impact evaluations and systematic reviews (SRs) by specific coverage and health outcomes. DPT, diphtheria, tetanus and pertussis vaccine; IPV, inactivated polio vaccine; BCG, Bacillus Calmette-Guérin vaccine; OPV, oral polio vaccine.

immunisation coverage, as opposed to other changes produced by the intervention.

## Coverage of intervention–outcome combinations

Commonly examined interventions such as caregiver incentives and motivation and health system governance, policies and financing are frequently evaluated for their effect on vaccination coverage outcomes. In comparison, most interventions have occasionally been evaluated for their effects of intermediate outcomes that precede coverage in the theoretical chain: behavioural, social and practical barriers faced by caregivers and communities or the health system's capacity to deliver vaccination services. Some exceptions are (1) sustained sensitisation and education campaigns, which are frequently evaluated for their effect on knowledge of immunisation and (2) formal HW training and education interventions, which have been evaluated for their effects on HW motivation and capacity.

## Cross-cutting themes and attention to equity

We examined included studies for the presence of evidence about themes that cut across our intervention and outcome framework. Many of these themes affect whether the study provides evidence about the effects of an intervention on particular vulnerable populations, either via subgroup analysis or by evaluating an intervention specifically targeted to that population.

Evidence patterns are quite similar across equity themes, with attention to each of these themes being relatively rare in IEs (figure 10). The most commonly measured is socioeconomic status (SES): 35 IEs (15%) address this theme through targeted interventions or subgroup analysis. The least commonly studied is hard-to-reach populations, which appears in 21 IEs (9%).

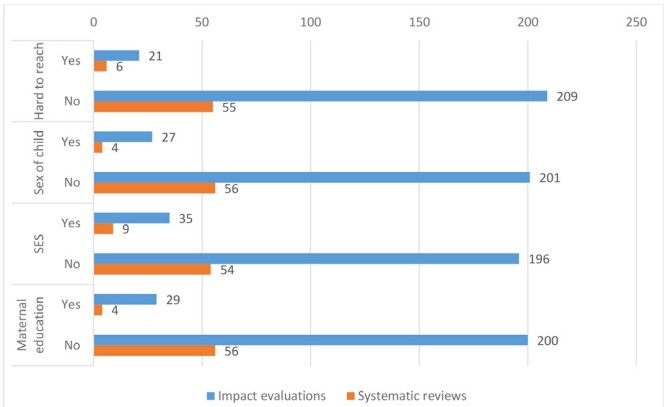

**Figure 10** Attention to equity in included studies. SES, socioeconomic status.

## Cost information

We recorded whether included studies provided any sort of information about the costs of the intervention, including any form of cost-effectiveness or cost–benefit analysis. Including such cost information is relatively rare: only 37 IEs (16%) and 9 SRs (15%) provide any such information.

## Confidence in findings of included SRs

About two-thirds of the 58 included reviews (39 SRs) received 'low' confidence ratings, with the remainder split roughly equally between 'medium' (9 SRs) and 'high' (10 SRs) confidence ratings.

## DISCUSSION: WHAT ARE THE EVIDENCE GAPS?

Most EGMs serve two primary purposes.[16] First, they assemble the evidence on a topic and provide an easy way for users to identify the evidence most relevant to them (ie, those studies examining interventions and outcomes of interest, or in particular regions or countries). Second, they identify gaps in the current knowledge base and a way to inform the decisions of researchers and research funders as to where they should direct their efforts and resources.

Reflecting the first purpose, this map can serve as a tool for policy-makers and practitioners working in the routine immunisation space to find rigorous evidence to inform decisions about which programmes to implement and how to implement them. High-quality syntheses of rigorous evidence are often the best sources of information on the effectiveness of interventions available to policy-makers and practitioners. Therefore, we recommend consulting the high-confidence syntheses identified in this EGM to inform the selection, design and delivery of interventions to improve immunisation outcomes (see reference list in online supplemental appendix C for the breakdown of included SRs by confidence level).

Reflecting the second purpose of an EGM, this map identifies several gaps that will need to be filled by the research community in order to have a robust evidence

base for decision-making on routine immunisation in LMICs.

A first gap is that evidence is very unevenly distributed geographically, with studies heavily concentrated in a handful of countries in sub-Saharan Africa and South Asia. From a policy and programming perspective, it is especially important to have evidence about what interventions are effective in places with the greatest challenges in delivering routine immunisation to their populations. We observe a number of gaps between where evidence is available and where it is most needed (see figure 5). This suggests a need for researchers and research funders to partner with implementers in these challenging contexts to conduct rigorous evaluations to identify what works to improve routine immunisation outcomes in these contexts.

Among interventions in our framework, there are major gaps in terms of outreach to certain groups that are less likely to be vaccinated, including migrants, refugees and vaccine-hesitant groups. While we have identified a concentration of evidence on use of community dialogue for increasing immunisation coverage, the amount of evidence on use of community resources, such as local leaders or selected community groups, to promote vaccination is relatively small. With community engagement approaches featuring prominently in the Global Vaccine Action Plan,[20] it will be important to further strengthen this evidence base for better guidance on programming and policy.

Another major gap concerns intermediate outcomes related to (a) behavioural, social and practical barriers faced by caregivers or (b) outcomes influencing the delivery of vaccination services such as the supply and capacity of formal HWs, supply chain management and the use of immunisation data in monitoring and planning. There is likely additional evidence on these intermediate outcomes, but our EGM suggests that it is relatively rare for rigorous quantitative studies to assess them in the specific context of promoting vaccination coverage.

In general, our EGM did not identify any glaring synthesis gaps in the form of large clusters of IEs on similar topics but no medium-confidence or high-confidence SRs synthesising the results of these evaluations. However, we note that there are 44 IEs on health provider incentives and motivation interventions, but only one high-confidence SR, which is now nearly a decade old.

Among SRs in our sample, the majority (67%) received a confidence rating of 'low' based on their methods. The SRs in our sample frequently lacked comprehensive search strategies and appropriate reporting of evidence, particularly with respect to the risk of bias of included primary studies (online supplemental appendix D, figures D.2 and D.3). Recognised guidelines for best practices in evidence synthesis, such as those provided by Cochrane[21] can guide researchers in avoiding bias in the identification of studies and the interpretation of results.

## Limitations

Our approach to coding multicomponent interventions presents some limitations. First, it risks overstating the volume of available evidence because the same study can appear in multiple rows in the map. Second, it can potentially distort theories of change. This is because, from looking at the map, it can appear that there is rigorous evidence about the effect of an intervention on an outcome, whereas in actuality there is only evidence about how that intervention affects the outcome as a component in a multicomponent intervention. To address this limitation, we provided an alternative analysis wherein all multicomponent interventions are classified under a single 'multicomponent' category (online supplemental appendix D, figure D.1).

Furthermore, it should be emphasised that our search strategy might have missed studies focused on intermediate outcomes, if assessing impact on vaccination-related outcomes was not a central component of the study. Therefore, these data should not be interpreted as showing, for example, that there is virtually no evidence about how to strengthen capacity among health service administrators. Rather, our EGM suggests that this outcome is rarely considered in the context of vaccination specifically.

## CONCLUSIONS AND IMPLICATIONS

Our identification of evidence gaps can provide direction for researchers and funders of research about high-priority areas for future investigation. The most glaring gap concerns intermediate outcomes on the capacity of the health system to deliver vaccination services, and on behavioural, social and practical barriers faced by caregivers. As noted above, our EGM results do not necessarily suggest an absolute dearth of evidence on health system strengthening. Rather, we find a lack of specific evidence tying health system capacity to the delivery of vaccination services. Thus, going forward researchers may wish to consider assessing the extent to which improvements in health system capacity lead to improvements in immunisation coverage, by including both as outcomes in evaluations. Specific areas requiring additional attention include: caregivers' attitudes about health providers, the social and practical factors influencing caregivers' decisions about vaccination and their access to vaccination services, the supply and capacity of formal HWs and administrators responsible for delivering vaccination services, use of data in monitoring and improving vaccination services and coverage of IPV.

Future research should pay greater attention to equity in immunisation studies—as emphasised in WHO's Immunization Agenda 2030[19]—by considering vulnerable groups in intervention and research design. These groups include female children, zero-dose children, caregivers with low levels of formal education, and households that are of low SES, are in hard-to-reach areas or are mobile and displaced. Furthermore, it is important

for future research to incorporate analysis not only of the effects of interventions, but their costs as well.

High-quality syntheses of rigorous evidence are often the best sources of information available to policy-makers and practitioners on the effectiveness of interventions. Therefore, we recommend consulting the high-confidence syntheses identified in this EGM to inform the selection, design and delivery of interventions to improve immunisation outcomes (see the list of included studies in online supplemental appendix C for the breakdown of included SRs by confidence level).

Given the current limitations of the evidence on health system outcomes in the context of immunisation, policy-makers seeking to strengthen health systems for the purpose of improving immunisation outcomes may wish to consult the broader literature on health system strengthening. While there is limited evidence linking improved health system capacity to improved vaccination coverage, robust theories of change suggest that adequate health system capacity is necessary, though not sufficient, for improving immunisation outcomes.

Finally, there are steps that policy-makers and practitioners can take when needing to make decisions regarding an intervention or outcome with inadequate evidence. For areas where there is a synthesis gap (ie, there are IEs available but no reliable SRs), one can consult low-confidence SRs (with appropriate caution) and individual IEs, although it is important not to place too much weight on any individual study. For those deciding to launch an intervention where relevant IE evidence is lacking, we suggest considering whether incorporating an IE of the programme would be feasible.

**Acknowledgements** We are grateful to John Eyers for valuable assistance with the development and execution of the search strategy. We also wish to thank the following people who supported this evidence gap map through excellent research assistance in the form of screening, data extraction or critical appraisals of systematic reviews: Himani Aggarwal, David Atika, Ashton Baafi, Beáta Berkovics, Malvya Chintakindi, Josh Furgeson, Suchi Kapoor, Lina Khan, Meital Kupfer, Harini Narayanan, Davi Romão, Agrima Sahore and Mansi Wadhwa. We thank Frédéric Cochinard for assistance in creating the online version of the evidence gap map. Finally, we are grateful to Molly Abbruzzese, Bidisha Barooah, Josh Furgeson, Doug Glandon, Mohamed Jalloh, Lisa Menning, Ada Sonnenfeld and three *BMJ Open* reviewers for helpful comments on a draft of this report.

**Contributors** ME, MJ and AB designed the research and supervised the screening and coding of articles to create the final dataset. ME conducted the analysis with input from MJ, AB and SSP. ME, MJ, AB and SSP wrote the manuscript. ME is the guarantor responsible for the overall content.

**Funding** This research was funded through a grant from the Bill & Melinda Gates Foundation (OPP1115129) to the International Initiative for Impact Evaluation, by which all the authors are employed as staff members or consultants.

**Competing interests** None declared.

**Patient and public involvement** Patients and/or the public were not involved in the design, or conduct, or reporting, or dissemination plans of this research.

**Patient consent for publication** Not applicable.

**Ethics approval** Because this research does not involve human or animal subjects, we did not secure external ethical review or approval.

**Provenance and peer review** Not commissioned; externally peer reviewed.

**Data availability statement** Data are available upon reasonable request. Data can be accessed via the Dryad data repository at http://datadryad.org/ with the doi: 10.5061/dryad.41ns1rnhr.

**ORCID iDs**
Mark Engelbert http://orcid.org/0000-0002-3665-1257
Monica Jain http://orcid.org/0000-0001-5428-377X
Shradha S Parsekar http://orcid.org/0000-0002-8824-9198

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
