## [Reviewer comments · BMJ Open]

ARTICLE DETAILS

TITLE (PROVISIONAL)	Improving routine childhood immunisation outcomes in low- and middle-income countries: an evidence gap map
AUTHORS	Engelbert, Mark; Jain, Monica; Bagai, Avantika; Parsekar, Shradha

VERSION 1 – REVIEW

REVIEWER	Oyo-Ita, Angela University of Calabar, Nigeria
REVIEW RETURNED	23-Nov-2021

GENERAL COMMENTS	This is a well written paper that addresses the question on what evidence is available in low and middle-income countries to improve vaccine uptake. The findings are useful to policy-makers and researchers.
--

REVIEWER	Biundo, Eliana GSK Belgium
REVIEW RETURNED	12-Jan-2022

GENERAL COMMENTS	General comments: - the paper is very descriptive in terms of frequency of themes covered in the reviewed articles but fails to make a link between intervention type and outcomes, nor to report which of the mapped interventions were successful and why. It is useful to assess where evidence gaps are but at the same time the results of this study would not help to inform a policy maker. My suggestion is to analyse more in detail studies that have outcomes of the intervention and as a minimum describe it. Having done similar analysis in the past I am aware often studies do not provide details of the outcomes as they might materialize later in time, but I would expect such a study to report any relevant evidence on the topic.- Discussion section is very short and fails to elaborate on some of the implications of the findings and again, should make at least an attempt to describe successful interventions. - Figure 2 - the second level intervention 'Education and Training' has the coloring of caregiver-oriented while in the Appendix (pag 5) it belongs to health system-oriented. - section 4.3 (line 22) - it is interesting to explore these themes however it comes as a surprise to the reader as it was not mentioned in the methods section.- Discussion section - the section is very short and fails to reflect on what the study findings mean for informing policy making. In particular there is no reflection on the link between type of intervention and outcomes.
---

REVIEWER	Wariri, Oghenebrume LSHTM
REVIEW RETURNED	27-Feb-2022

GENERAL COMMENTS	In this study, Engelbert et al. created an evidence gap map (EGM) to characterise the evidence base on the effectiveness of interventions in improving routine childhood immunisation outcomes in low- and middle-income countries. To draw their evidence, they searched several databases and grey literature sources for relevant impact evaluations and systematic reviews. This is a very important and timely study as we now envision a world where everyone, everywhere, at every age, fully benefits from vaccines, through the IA2030. I believe that evidence from studies like this will be important to guide the direction of future research and immunisation policy. It is obvious that a huge amount of work has gone into gathering and analysing the data presented in the manuscript. The manuscript is well written and I do not have any major issues. That notwithstanding, there are some other issues that the authors should consider addressing to improve the overall quality of the manuscript. I have annotated them below and made some suggestions on how to address them.  1. L&MICs should be written as LMICs 2. Consistency with the use of abbreviated form and the full format. For example, repeated use of IEs (Impact Evaluations) and SRs (Systematic Reviews) even after they have been defined 3. It is my opinion that section 2.2, i.e., the inclusion and exclusion criteria should be moved to the 'Methods' section following the IMRaD (Introduction, Methods, Results and Discussion) framework. 4. The first paragraph on page 9 (lines 3-10) sounds more like material suited for the Methods section rather than in the Results section as it is not describing the findings from this study. The same can be said of the 3rd paragraph on the same page (lines 27-39) where the authors describe their outcome taxonomy, rather than 'Results'. 5. The second paragraph on page 10, lines 22-29 sounds like a limitation of the study, thus, the authors should consider moving it to the discussion section of the manuscript. I thank the Editors for the opportunity to review this important work.
---

VERSION 1 – AUTHOR RESPONSE

Reviewer: 1

Dr. Angela Oyo-Ita, University of Calabar, Nigeria Comments to the Author:

This is a well written paper that addresses the question on what evidence is available in low and middle-income countries to improve vaccine uptake. The findings are useful to policy-makers and researchers.

Reply: We thank Dr Oyo-Ita for providing a review and for the positive words about our manuscript.

Reviewer: 2

Dr. Eliana Biundo, GSK Belgium

Comments to the Author:

General comments:

2.1. the paper is very descriptive in terms of frequency of themes covered in the reviewed articles but fails to make a link between intervention type and outcomes, nor to report which of the mapped interventions were successful and why. It is useful to assess where evidence gaps are but at the same time the results of this study would not help to inform a policy maker. My suggestion is to analyse more in detail studies that have outcomes of the intervention and as a minimum describe it. Having done similar analysis in the past I am aware often studies do not provide details of the outcomes as they might materialize later in time, but I would expect such a study to report any relevant evidence on the topic.

Commented [SP1]: Mark and Monica

Reply: We thank the reviewer for this comment and agree that information about which interventions are effective is essential for policy decision-making. However, such information is beyond the scope of an evidence gap map, where the aim is to map the evidence so that users of the map can readily identify any studies speaking to these effectiveness questions, including existing systematic reviews.

Questions relating to intervention effectiveness are best addressed through systematic reviews—indeed, one of the aims of an EGM is to identify “synthesis gaps” where an SR is needed. We also note that we address the effectiveness of one type of intervention (namely, community-engagement strategies for increasing routine immunisation) in an SR (forthcoming in *Campbell Systematic Reviews*) that is a companion to this EGM.

2.2. Discussion section is very short and fails to elaborate on some of the implications of the findings and again, should make at least an attempt to describe successful interventions.

Reply: Thank you for pointing this out. We have expanded the discussion section to start by explaining how EGMs can support decision-making in general (by directing people to relevant evidence and by informing priority areas for further research). We then highlight the frequent usefulness of high-confidence SRs for decision-making and recommend that policymakers using our map look there first for evidence. We then discuss several evidence gaps and how they might be filled by researchers and research funders.

2.3. Figure 2 - the second level intervention 'Education and Training' has the coloring of caregiver-oriented while in the Appendix (pag 5) it belongs to health system-oriented.

Reply: Thank you pointing out this error. Yes, it belongs to health system-oriented category; we have accordingly made the changes in figure 2.

2.4. section 4.3 (line 22) - it is interesting to explore these themes however it comes as a surprise to the reader as it was not mentioned in the methods section.

Reply: Equity themes—such as whether studies conducted and reported subgroup analyses based on socio-economic characteristics or on hard-to-reach populations—were explored as a part of the seven cross-cutting themes included in the intervention outcome matrix. These themes have been explicitly mentioned in the methods section as suggested (p. 6).

2.5. Discussion section - the section is very short and fails to reflect on what the study findings mean for informing policy making. In particular there is no reflection on the link between type of intervention and outcomes.

Reply: As noted in the reply to 2.2, we have expanded the discussion section. We have also added a section (4.3) on trends and gaps in the evidence on particular combinations of interventions and outcomes.

Reviewer: 3

Dr. Oghenebrume Wariri, LSHTM

Comments to the Author:

In this study, Engelbert et al. created an evidence gap map (EGM) to characterise the evidence base on the effectiveness of interventions in improving routine childhood immunisation outcomes in low- and middle-income countries. To draw their evidence, they searched several databases and grey literature sources for relevant impact evaluations and systematic reviews.

This is a very important and timely study as we now envision a world where everyone, everywhere, at every age, fully benefits from vaccines, through the IA2030. I believe that evidence from studies like this will be important to guide the direction of future research and immunisation policy.

It is obvious that a huge amount of work has gone into gathering and analysing the data presented in the manuscript. The manuscript is well written and I do not have any major issues. That notwithstanding, there are some other issues that the authors should consider addressing to improve the overall quality of the manuscript. I have annotated them below and made some suggestions on how to address them.

3.1. L&MICs should be written as LMICs

Reply: We originally chose “L&MICs” to make clear this is an abbreviation for “low- and middle-income countries”, rather than “lower-middle-income countries”, which is also sometimes abbreviated “LMICs”. However, as the reviewer notes, “LMICs” is a commonly accepted abbreviation, so we have followed the suggestion and replaced L&MICs with LMICs in the entire document.

3.2. Consistency with the use of abbreviated form and the full format. For example, repeated use of IEs (Impact Evaluations) and SRs (Systematic Reviews) even after they have been defined

Reply: We have made the changes as suggested, however, for readability we preferred to retain the full format in our description of objectives and in (sub)headings.

3.3. It is my opinion that section 2.2, i.e., the inclusion and exclusion criteria should be moved to the ‘Methods’ section following the IMRaD (Introduction, Methods, Results and Discussion) framework.

Reply: As suggested, ‘inclusion and exclusion criteria’ (formerly section 2.2., now section 3.1) has been moved to Methods.

3.4. The first paragraph on page 9 (lines 3-10) sounds more like material suited for the Methods section rather than in the Results section as it is not describing the findings from this study. The same can be said of the 3rd paragraph on the same page (lines 27-39) where the authors describe their outcome taxonomy, rather than ‘Results’.

Reply: We have re-organized the information presented in the paragraphs identified by the reviewer. This information can now be found in the methods section (p. 6).

3.5. The second paragraph on page 10, lines 22-29 sounds like a limitation of the study, thus, the authors should consider moving it to the discussion section of the manuscript.

Reply: We have moved the suggested paragraph to the section on limitations of the study (p. 14).